# Applications of Molecular Dynamics Simulation in Protein Study

**DOI:** 10.3390/membranes12090844

**Published:** 2022-08-29

**Authors:** Siddharth Sinha, Benjamin Tam, San Ming Wang

**Affiliations:** MoE Frontiers Science Center for Precision Oncology, Cancer Center and Institute of Translational Medicine, Faculty of Health Sciences, University of Macau, Macau SAR, China

**Keywords:** molecular dynamics simulations, enhanced sampling techniques, membrane dynamics, GPCRs, lipid-protein interactions, ACE-2 membrane receptor

## Abstract

Molecular Dynamics (MD) Simulations is increasingly used as a powerful tool to study protein structure-related questions. Starting from the early simulation study on the photoisomerization in rhodopsin in 1976, MD Simulations has been used to study protein function, protein stability, protein–protein interaction, enzymatic reactions and drug–protein interactions, and membrane proteins. In this review, we provide a brief review for the history of MD Simulations application and the current status of MD Simulations applications in protein studies.

## 1. Introduction

The essence of Molecular Simulations (MS) is a statistical mechanics and numerical method governed by the Newtonian laws of motion [1] for molecular properties, i.e., velocity, position, and energy, towards insights of molecular system while retaining macro-system physio-chemical properties. Two factors have promoted the increased application of molecular simulations over the years (Figure 1). One is the growing availability of experimentally determined protein structures, such as membrane proteins (ion channels, neurotransmitters and GPCRs etc.) [2,3], the other is the wide availability of graphics processing units (GPUs), which allows running simulations locally. MS typically analyses protein structure at a minimum of nano to micro-second time scale to reveal the dynamic nature of protein molecules covering a wide variety of biomolecular processes, such as conformational change, ligand binding and protein folding. Among the numerous approaches to MS, the Monte Carlo (MC) Simulation sampling method and the MD Simulation method are the two common methods. The basic concept of MCS is to generate an ensemble of conformation under specific thermodynamics conditions through stochastic approach; whereas the concept of MD Simulation is to iterate a time-dependent Newtonian equation of motions for hard sphere particles in a system [4,5], which can provide an ensemble of thermodynamic properties. 

## 2. A Brief History of Molecular Simulations

MS was first introduced in 1949 by Metropolis et al. to study particle interaction [6]. Metropolis proposed a probabilistic approach to approximate the “properties” of a set of particles [6]. Instead of treating particles as individuals, simulation was applied to measure the interactions of all particles until they reach equilibrium by the governing laws. Its success inspired the development of MS by Alder and Wainwright in 1959 [7]. The early MS algorithm used a rudimentary electronic computer to iterate atom collision. Each atom was assigned an initial velocity and position. Based on the elastic collision, the MS algorithm was applied to simulate attraction and repulsion of particles. In 1964, Rahman et al. published the first study in using MS to analyze liquid Argon [8]. The work demonstrated that MS was indeed possible to analyze Lennard Jones potential for interactions between Argon atoms. In 1971, Rahman and Stilinger reported their MS study on modelling liquid water, a system composed of molecules not just atoms [9]. Their work demonstrated that differing from its solid phases structure, liquid water consists of a random network of hydrogen bonds. In 1976, Warshel and Levitt expanded MS by integrating quantum mechanics and molecular mechanics (QM/MM) to study lysozyme reaction by proposing the exchange of the classical charge of atom *i* and *j* with quantum mechanics calculations [10]. In 1977, Karplus and collaborators first used MS to study protein by using constraint method to freeze out fast-degree freedom to reach longer simulation time [11,12]. Their study led to the Noble Prize in Chemistry awarded to Warshel, Levitt and Karplus in 2013 for the development of multiscale models for complex chemical systems [10]. Anderson et al. in 1980 used MS to sample the isoenthalpic (constant pressure) ensemble. Anderson’s solution to achieve constant pressure in MD Simulation sampling was to extend dynamic variable by including volume [13]. Parrinello and Rahman showed that the scheme can be generalized to include shape and volume fluctuations by using Lagrangian mechanics. This made it possible to study the issues such as crystallization and solid–solid phase transition [14]. Their idea of extending the system dynamic variables was to assume that the system exchanges energy with a fictitious pressure or temperature reservoir. Their method took into consideration the dielectric effect caused by the atomic polarizability and increased the accuracy of the binding site. In 1985, Car and Parrinello pioneered a scheme of combining MS with direct calculation of electronic structure by means of Density Function Theory (DFT). This work was important as it indicated the possibility of combining finite temperature into simulation for electronic structure calculations, which was not possible before [15]. During 1980s and 1990s, MS approach witnessed a rise in studies of condensed matter with growing access of enhanced computing power; further leading to the challenges of phase equilibria. Moreover, to address these challenges Panagiotopolus revised the MC algorithm, known as *Gibbs ensemble Monte Carlo,* to distinguish the phase equilibria approach that only require to simulate the involved phases but by-pass the interface [16]. Novel algorithms such as *blue moon ensemble* [17] hyper-MD [18] as well as advanced theoretical methods such as Nudged-Elastic Band [19] and String [20] were devised to address the challenges of time-scales (long-time dynamics of protein folding) and rare events. Further, the advancement in quantum programs outside chemistry field and the Noble prize in Chemistry 1998 being divided equally between Walter Kohn “for his development of density-function theory” and John A. Pople “for his development of computational methods in quantum chemistry” led to form a unified approach for molecular dynamics and density-function theory. Over the following years, time-dependent density-function theory (TDDFT) further enhanced the accuracy of large-scale simulations of excited state dynamics [21,22,23]. TDDFT-MD coupled simulations to simulate excited state dynamics of biomolecules and other nanostructures achieves high accuracy through utilizing small number of basic function thereby significantly reduced the memory requirements and computation time compared to plane-wave and real-space grid bases [24]. Furthermore, utilizing multiple computer processors in parallel for MD force calculations substantially enhanced with IBM’s Blue Matter code for its Blue Gene/L general-purpose supercomputer [25], resulting in improved parallel performances for the widely used MD platforms NAMD [26] GROMACS [27] AMBER [28]. Increasing innovation and with advent of GPU (Graphics processing units) and special-purpose processors such as Anton (parallel supercomputer to enable fast MD simulations) having computing power to perform up to 20 μs/day [29] further accelerated the simulation study in different biochemical processes. However, long-timescale simulations requires stringent force field (discussed in following section) compared with short-timescale simulations. To conclude this brief history of MS, it would be appropriate to remark that MS has clearly established itself as a key scientific instrument driven by enhanced computing power, fast and efficient algorithms and force fields (FF) are demonstrated by growing number of publications utilizing both experiments and simulation tools. Major breakthroughs over the years in MS studies are shown in Figure 2.

## 3. Basic Concept of Force Field

Currently, it is a routine to simulate proteins with hundreds of amino acid residues at 10–100 ns surrounded by water and salt [30,31,32]. User-friendly platforms are widely available, i.e., GROMACS [33], AMBER [28], vCHARMM [34], DL_POLY [35], NAMD [26], LAMMPS [36] have been developed for MD Simulations analysis. The output of the platforms can be visualized and analyzed by external software, i.e., VMD [37], Chimera [38]. However, robust simulation requires appropriate parameters for studying a physical system. Force field, a set of mathematical expressions and parameters to describe the inter- and intra- molecular forces, are also essential to describe a physical system.

Three major molecular models have been developed: all-atom [39,40], coarse grained (CG) [41,42] and all-atom/coarse-grain mixed models [43,44,45] (Table 1). The all-atom force field for MD Simulation of lipid bilayers includes CHARMM, AMBER and OPLS-AA. GROMOS is an atomistic force field with an exception such as CH_n_ modelled as united-atoms [46]. CHARMM (Chemistry at HARvard Macromolecular Mechanics) forcefield for lipids is widely used for simulating lipid bilayer and membrane proteins [47,48]. CHARMM force field is continuously updating and improving with the most recent version of CHARMM36m [49]. CHARMM36 lipid forcefield is parameterized for lipids [39], CHARMM36 DNA and CHARMM36 RNA are parameterized for DNA and RNA [50,51], CHARMM36m is parameterized for protein, and CHARMM General Force Field (CGenFF) is parameterized for drugs and general usage [52]. AMBER (Assisted Model Building with Energy Refinement) forcefield was developed in parallel. It treats all hydrogen atoms explicitly as CHARMM [53]. AMBER was designed and parameterized for specific biological systems: AMBER lipids 21 was parameterized for lipids [54]; AMBERff19SB was parameterized for proteins [55]; AMBER OL15 and AMBER OL3 were parameterized for DNA and RNA [56,57]; General AMBER forcefield (GAFF) was parameterized for drugs and general usage [58]; OPLS-AA (Optimized Parameters for Liquid Simulations All Atom) [59] was initially designed for simulating thermo-dynamical properties of short-chain hydrocarbons alkanes and later expanded to include lipids through a parameter set called OPLS/L [60], although the availability of lipids in the OPLS/L forcefield has not been as diverse as that of CHARMM and AMBER-compatible force fields. The latest improvement of OPLS-AA/M was its modification for peptides and protein torsional energetics [61]. The GROningen Molecular Simulation (GROMOS) forcefield utilizes a different approach for simulating analysis by fitting the parameters against experimental thermo-dynamic data. Its forcefield was generalized into a single package. The latest version is GROMOS 54A8 package updated in 2012 [62].

Compared to all-atom models, coarse-grained models significantly reduce the computing time by decreasing the number of particles explicitly during simulations. Over the last decade, coarse-grained model has also been widely used in protein [63] and nucleic acid studies [64,65]. Different coarse-grained models have been developed to extend the timescale of the simulation, since the first model used the concept of coarse grain in 1975 by Levitt and Warshal [66]. One of the most popular models is the MARTINI for membrane proteins [42], in which several atoms in protein and lipid are approximated as a single bead and four water molecules are treated as a single particle (known as one bead 4:1 mapping) although the beads can differ by their polarity or hydrophilicity. For particular cases, smaller beads can also be used, such as 3:1 and 2:1 mapping [67]. In MARTINI version 2.2, beads classified into 18 types are categorized into four groups: Q (charged), P (polar), N (intermediate) and C (apolar). In the latest version MARTINI 3, 29 beads have been sorted into seven groups with additional groups of halo-compounds (X), divalent ions (D) and water (W) [68]. MARTINI ELNEDIN model modified by utilizing an elastic network, with the peptide backbone beads position on the Cα atoms and heavier bead mass, improves the conformation transition in simulation [68]. MARTINI-Dry version provides an implicated solvation model [69]. The Born model is another model where the effects of the solvent and membrane are included implicitly in the simulation [70,71]. Implicit solvent forcefield is less used as it can cause significant errors due to it smoothen energy landscapes, which causes protein structure to deviate from the experimental crystal structure [72,73]. Coarse-grained protein models have been mainly used for analyzing protein folding mechanism and protein structure prediction [74,75]. Every alternate year, the CASP (Critical Assessment of Protein Structure Prediction) experiments provide an excellent platform to test the performance of coarse-grained models for predicting structures [76]. Several coarse-grained protein models apart from MARTINI are as follows: UNRES (united residue) [77], AWSEM (associated memory, water mediated, structure and energy model) [78], OPEP (optimized potential for efficient protein structure prediction) [79], SURPASS (Single United Residue per Pre-Averaged Secondary Structure fragment) [80] and CABS (C-alpha, c-beta, side chain) [81] models have been increasingly utilized for protein folding, structure prediction and interactions. PRIMO [82] and Scorpion [83] (solvated coarse-grained protein interaction) models are increasingly used in peptide and small protein structure prediction and protein–protein solvated complexes. The Rosetta centroid mode (CEN) model developed by Rohl et al. is also one of the widely used coarse-grained protein models in CASP protein structure prediction, de novo blind predictions, protein–protein and protein–ligand docking and modelling of protein-DNA interaction [84]. Coarse-grained models have been further utilized in nucleic acid molecular dynamics to analyze the three dimensional (3D) structural models of RNA [85,86,87]. Ding et al. introduced the discrete molecular dynamics (DMD) utilizing coarse-grained model to rapidly explore the conformational folding of RNA molecules [88]. Recently, Jonikas et al. have developed a fully automated coarse-grained model NAST (the nucleic acid simulation tool) using statistical potential capable enough to ensemble over 10,000 RNA plausible (3D) structures [89].

## 4. Molecular Simulations in Protein Study

The importance of MS arises from the fact that biomolecules such as proteins are under a dynamic state of motion, which is essential for the function of biomolecules. Although multiple experimental techniques can reveal the structural features of biomolecules, they are often incapable to show the dynamic features. MS provides a means to model the flexibility and conformational changes in the biomolecule at atomistic level, which is difficult to achieve by experimental approaches [11]. MS is more effective when combined with experiments to validate and improve the accuracy of experimental results. A key feature of MS is its ability to mimic both the in vitro and in vivo conditions, for example, at different pH conditions, in the presence of water and ions, at different salt or ionic concentrations, and in the presence of a lipid bilayer and other cellular components [92]. MS has been used to study multiple protein-related issues, such as protein-binding, protein–protein interaction and signaling [93]. The followings are examples.

### 4.1. Applications of Molecular Simulations in Membrane Proteins

MS has been increasingly applied in membrane protein analysis [94], such as membrane protein structure and organization, membrane protein permeability, lipid-protein interaction, protein–ligand interaction, protein structure and dynamics [95,96]. MS is also used in combination with a wide variety of experimental techniques to address protein structure-related questions, including X-ray crystallography, cryo-electron microscopy (cryo-EM), nuclear magnetic resonance (NMR), electron paramagnetic resonance (EPR) and Foster resonance energy transfer (FRET) [97]. For example, MS can minimize the gap between NMR structures and X-ray crystallography structures, allowing for better analysis of structural instability and interaction [98].

Membrane protein can be classified into three classes: integral, peripheral and lipid-anchored [99]. Based on the interaction of membrane protein with lipid bilayer, the three classes can be further divided into eight types: (1) type I membrane protein; (2) type II membrane protein; (3) type III membrane protein; (4) type IV membrane protein; (5) multipass transmembrane protein; (6) lipid chain-anchored membrane protein; (7) Glycosylphosphatidylinositol (GPI)-anchored membrane protein; and (8) peripheral membrane protein [99]. In a biological membrane, lipid molecules are arranged spontaneously to form a lipid bilayer having hydrophobic chains in the interior and hydrophilic groups at the exterior [100]. Membrane protein such as transporters, ion channels etc. plays significant roles in transportation of ions, polypeptides and other substrates through lipid bilayers [101]. Membrane receptor proteins responsible for signal transduction is also one of the important functions of membrane protein [102]. Compared with soluble proteins, determination of the structure for membrane proteins using X-ray, NMR and cryo-EM is more challenging, and the number of membrane protein structures in protein databases, i.e., PDB, JenaLib, OPM [103,104,105] is also limited [106,107]. Furthermore, as membrane proteins often undergo large conformational changes, a single structure is not sufficient to understand the mechanism of their biological function. Therefore, increasing attention has been paid in applying simulations to study membrane proteins. The structures of many membrane proteins have been experimentally determined, e.g., many ion channels, neurotransmitters, transporters and G protein-coupled receptors (GPCRs) etc., the information facilitate simulation study. Furthermore, the increased power and accessibility of MD Simulation by computer hardware, particularly GPU (graphical processing unit), allows simulations to be run locally at modest cost [108,109,110]. Nowadays, simulation is often applied in the timescale of microseconds, thus making it possible to trace biological events from the early studies, which primarily focused on phospholipid bilayers such as DPPC (dipalmitoylphosphatidylcholine) or DMPC (dimyristoylphosphatidylcholine) [40,111,112]. To simulate various biological phenomena such as aggregation, large conformational changes and membrane protein folding, Hensman and Okamoto first applied the enhanced conformational sampling method [113,114,115,116,117]. They compared the accuracy and efficiency of different molecular models in glycoprotein A (GpA), phospholamban (PLN), amyloid precursor protein (APP) and mixed lipid bilayers [118], and observed that the predicted GpA, PLN and APP structures using the replica-exchange MD (REMD) and replica-exchange umbrella sampling (REUS) approaches are comparable with the data from experiments, suggesting that the model and simulation approaches are sufficiently accurate.

### 4.2. Simulations of Integral Membrane Protein (GPCRs)

G protein-coupled receptors (GPCRs) are internal membrane proteins (IMPs) consisting of 7-transmembrane helix. They are the largest membrane receptors. There are about 800 GPCRs identified in the human genome [119], over a quarter of drugs target GPCRs [120,121,122]. In 2020, 24 new drugs targeting 16 GPCRs have been clinically approved, and 44 new drugs targeting GPCRs were under 100 clinical trials [123]. Simulation studies have drastically helped improve understanding of GPCRs structures and functions [124,125]. Dahl and Weinstein (1990) pioneered the MD Simulations studies of GPCR on dopamine, serotonin and opioid receptors [126]. With the X-ray determined crystal β_2_AR structures [127], microsecond-long MD Simulation of β_2_AR reveal multiple cholesterol (lipid bilayer) interactions distributed unequally between the extracellular (EC) and intracellular (IC) sides with variable binding strength [128]. There are three key areas where MD Simulations provide unique insights into dynamic properties of GPCRs: the change in conformations that occur between different GPCR active and inactive states, interaction of GPCRs with ligand/inhibitors and effects of lipids on the conformational dynamics of GPCRs.

Dror and colleagues utilized long time-scale MD Simulations to identify key connector region that connects GPCR canonical binding sites to G-protein binding site [129], Moreover, the conformations of the G-protein were key determinant as the inactive G-protein binding site restricts the connector region (GPCRs) to its inactive conformation [129]. The study performed a total of 92 simulations for ~656 µs time period to analyze the mechanism for GPCRs transition from inactive to active state. Further, using similar protocol and Anton (a supercomputer designed for accelerating MD Simulations) [130], Schneider and colleagues performed MD Simulations to analyze the differences between full agonists (Morphine) and biased agonists (TRV-130) in mutual information networks for the µ opioid receptor active state (PDB: 5C1M) [131]. The results clearly indicated that biased inhibitors interact with smaller set of residues, thereby make it easy to analyze the binding pattern experimentally.

GPCRs represent a broad spectrum of drug targets as they have pivotal roles in many physiological functions (neurotransmitters, environmental stimulus, chemokines etc.) and in disease development including cancer and cancer metastasis [132]. GPCRs are particularly useful for drug discovery due to their ability to modulate a variety of intracellular signaling pathways, including the activation of G proteins and β-arrestins [133]. Identification of novel molecules targeting GPCRs face several challenges as these proteins exist in different conformations rather than a single inactive and activated state [125]. Recent studies have used long unbiased MD Simulations for ~ 50 µs to predict the binding poses of TRV-130 to the μ-opioid receptor (MOR) [131], the allosteric ligands to the M2 muscarinic receptor (M2) [134], and ML056 to the sphingosine-1-phosphate receptor 1 (S1P1R) [135]. Further, Marino et al. applied meta-dynamics to study the ligand binding to GPCRs to predict the binding pose of a PAM, BMS-986187 to the δ-opioid receptor (DOR) as well as to MOR (G protein agonist) [136]. Further, the need to develop new protocols to decrease the computational time and increase the performance of the algorithms, resulted in Supervised MD (SuMD) capable of reducing the total simulation time from microsecond to nanosecond timescale [137]. The SuMD protocol was applied to binding analysis of numerous ligands to the A_2A_ adenosine receptor, resulting in significantly reducing the simulation time, for example the analysis of ZM241385 (PDB: 3EML), T4G (PDB: 3UZA), T4E (PDB: 3UZC) reproduced the crystallographic pose in approx. 60ns, 65ns and 110ns, respectively [137]. MD Simulations can reveal specific GPCR residues and ligand–receptor interactions responsible for the allosteric transmission, based on dynamical information derived from the simulations.

Lipids (cholesterol, etc.) also play a role in the function of GPCRs in addition to ligands and ions [138,139]. Early studies utilizing classical MD Simulation of A_2A_ adenosine-bound receptor (PDB: 2YDO) resulted in identification of potential cholesterol sites in GPCRs [140] with three binding sites. The third binding site, especially, demonstrated the same binding pattern in alignment with X-ray crystallographic structure of same receptor (PDB: 4EIY) [141]. MD Simulations was also utilized to analyze the mechanism of other lipids (simple/mixed zwitterionic bilayers) modulating A_2A_ receptor structure [142]. Simulations for 0.25 ms revealed that the lipid bilayers had different effects on the stability of the active state of native receptor. Moreover, simulation studies revealed that phospholipids can compete with G-protein binding site, suggesting that lipid binding at intracellular end can hinder G-protein binding, leading to modulation of GPCRs by phospholipids [143]. Further, a GPCR database has been developed, with reference data and tools for both analysis and visualization [144,145].

### 4.3. Simulations of Interaction between SARS-CoV-2 Spike and Membrane ACE2 Receptor

The outbreak of COVID-19 caused by severe acute respiratory syndrome coronavirus-2 (SARS-CoV-2) is an example of showing how MD Simulations can be used to understand the relationship between SARS-CoV-2 and the human host. SARS-CoV-2 infects human cells through its spike (S) protein binding to the angiotensin-converting enzyme-2 (ACE-2) receptor in the human cell membrane. SARS-CoV-2 constantly mutated its spike to increase its infection to the host. New mutants including alpha, beta, gamma, delta and omicron strains have been generated [146] carrying L452R, T478K, E484K, E484Q, and N501Y mutations. A typical example is the delta mutant, which contains 10 mutations of T19R, G142D, 156del, 157del, R158G, L452R, T478K, D614G, P681R, D950N in its S protein, and the double mutation L452R/T478K is located in RBD [147] (Figure 3).

MD Simulations provides a powerful tool to reveal the structural and conformational basis of the new mutants to ACE2 [148]. Massive-scale MD Simulations using state of art supercomputer machines have been used to gain insights into the biology of SARS-CoV-2 [149]. Amaro et al. used ~250,000 processing cores and ~4000 processor nodes in their MD Simulations study [150]; their results showed that glycans play a significant role in S-protein binding [151]. Taiji et al. used a drug discovery supercomputer MD GRAPE-4A to analyze the structural dynamics of M^pro^ of SARS-CoV-2 [152]. Acharya et al. used a supercomputer “Summit” to perform MD Simulations on 8000 compounds to screen for potent inhibitors to S-protein and identified 77 small-molecule drug compounds [153]. Remarkably, the folding@home computing project involving over a million-citizen scientists performed an unprecedented 0.1 second MD Simulations to simulate SARS-CoV-2 [154], revealed how the S-protein uses conformational change to escape host immunity, and subsequently identified the hidden cryptic pockets that were extremely difficult to capture by experimental approaches.

We also applied MD Simulations to study the effects of SARS-CoV-2 mutations on RBD domain binding affinity with ACE2. We studied the mechanism of the increased transmissibility of SARS-CoV-2 variants with double RBD mutations [149] by investigating the changes in binding pattern and structural conformation between the ACE2 receptor and four SARS-CoV-2 variants containing three RBD double mutations of L452R/T478K (delta) [155], L452R/E484Q (kappa) [156] and E484K/N501Y (beta, gamma) [157,158]. We used a combinational approach in the study, including 3D-protein structure, protein–protein interaction, molecular dynamics simulation, superimposed protein structure, affinity binding, and antibody binding mapping. We observed that the N501Y caused mild structural change and increased the binding affinity of the S protein to ACE2 [159]. We also observed that the binding energy of N501Y variants increased to –48.92 kcal mol^−1^, consistent with the observations by other in vitro studies showing the binding of Y501 increased 10-fold gain of binding affinity and in vivo studies showing N501Y imparted cross-species transmission [160,161,162]. E484 has a positive (opposed) binding affinity with the ACE2, but the variant K484 has significantly increased its binding affinity with the ACE2 [163]. This indirectly changed RBD structure configuration and strengthen other key binding residues (i.e., Y505, F486) in the RBM during the spike protein approaching the ACE2, leading to the increased binding affinity [164,165]. The substitution of K, Q, or P residues at the E484 position was identified and these variants assisted the virus to escape host immune defenses [166]. E484K mutation caused a 50% loss of neutralizing activities by antibodies, and a 3 to 6-fold reduction in neutralization by sera of the individuals who received mRNA-vaccine. Simulation with 26 common antibodies found in humans showed that up to 85% showed weaker binding affinities to the E484K mutated strain [167]. Double-mutation in the beta and gamma strains increased the binding strength of RBD as they changed the energy landscape of the RBD by ~25%. The combination of E484K immune escape capabilities and N501Y increased the binding affinity, causing ~50% higher transmissibility [168]. Our study showed that the three double mutated RBD all alter the wildtype RBD structure in the ways much different from those caused by the RBD single mutations, enhanced the binding of the mutated RBD to ACE2 receptor, changed antibody binding, leading to the increased infection of SARS-CoV-2 to the host cells (Figure 4).

## 5. Challenges and Future Opportunities

Increments and leaps of improvements are continuously produced by many research groups and developing new solutions for various persistent challenges remain the focus of research. At present, managing enormous information generated by simulations with every molecule represented in atomistic detail is a big challenge. Currently, it is impractical to share the primary data as there is no MD Simulations database [169,170,171,172]. Possible solutions include reducing the data size by using snapshots at different time points of simulation and removing insignificant parts of the system such as solvent, and maintaining the full dataset but allowing remote analysis so that only the results instead of the actual dataset need to be transmitted [173].

The connection between experiments and simulations is an important step complementary to each other [173,174]. This can be further enhanced through improving the FF or the method of simulations. For example, simulation accuracy can be significantly improved by integrating quantum mechanics on-the-fly simulation. However, many theoretical challenges and long computational time may prohibit the merging of quantum mechanics with molecular mechanics. The progress in this area remains stagnated since the 2000s mainly due to the number of electrons (i.e., number of basis sets to represent electronic wave function) involved in a system, integration of time, and calculation of the system beyond ground state [175]. To circumvent such problem, polarizable FF were developed to approximate dielectric effects in MD Simulations. However, there is a need to develop a polarizable FF for better accuracy than current versions of FF (AMBER99SB-ILDN, CHARMM22-CMA, GROMOS, OPLS-AA) in order to study protein in multi-scale environments [174].

Another key area of research gaining momentum is the integration of machine learning (ML) and deep learning (DL) techniques into MD Simulations. The incorporation brought various significant new research directions to analyze protein trajectories and protein structures. ML and DL can analyze non-linear complex systems by recognizing regular and similar patterns in the data. In particular, substantial expansion has been made that ML and DL utilize to create an adaptive force field on-the-fly [176,177,178], increasing the simulation timescale [179], and protein–protein/protein–ligand interactions [180,181]. ML and DL are becoming new potential tools for analyzing large amounts of data produced by MD Simulations.

Currently, it remains a challenge for researchers without high-end computing backgrounds to use MS to study the system of their interest. A user-friendly interface such as automation in MD Simulations need to develop. One such remarkable example was made by P. Arantes et al. (2021). They presented a user-friendly front-end running MD Simulations system using openMM toolkit on the Google colab framework [182] and cloud-computing scheme for performing MD Simualtions on microsecond time scale. Regardless the challenges, MD analysis is becoming a mainstream tool in basic and applied biology.

## Figures and Tables

**Figure 1 membranes-12-00844-f001:**
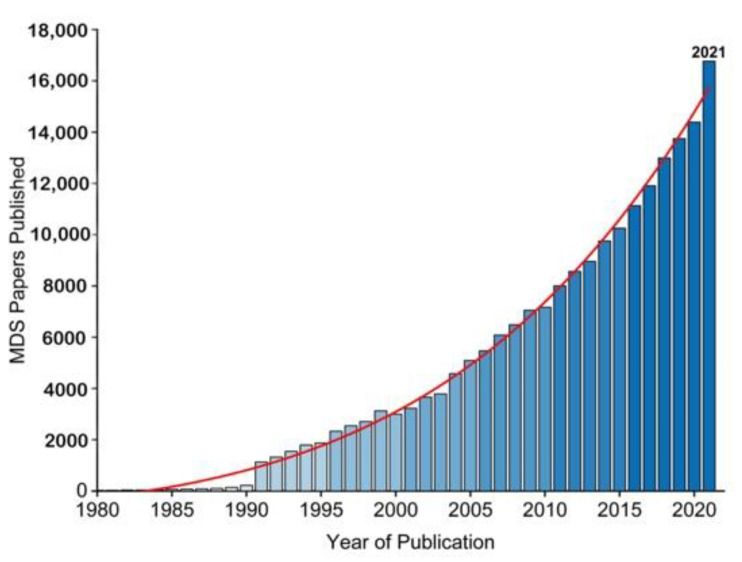
The growing use of MD Simulation studies over the years as reflected by publication (1980–2021). Data was from Web of Science.

**Figure 2 membranes-12-00844-f002:**
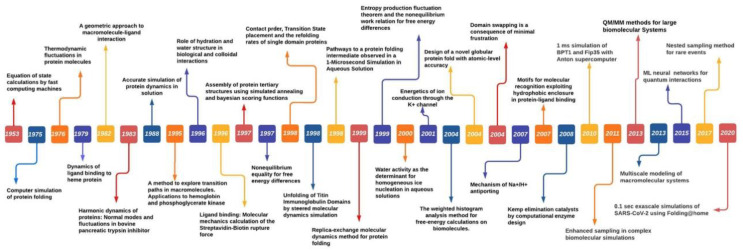
The Molecular Simulations timeline showing the breakthrough achievements in MD Simulation studies.

**Figure 3 membranes-12-00844-f003:**
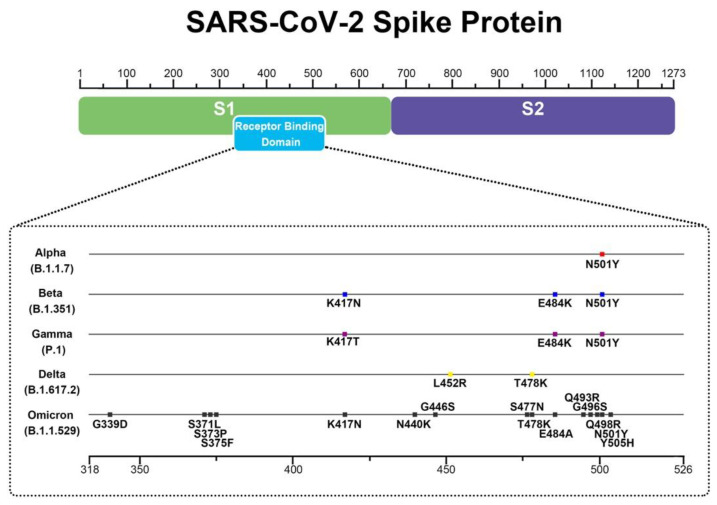
The RBD domain of SARS-CoV-2 spike protein showing the mutations in Alpha, Beta, Gamma, Delta and Omicron mutants.

**Figure 4 membranes-12-00844-f004:**
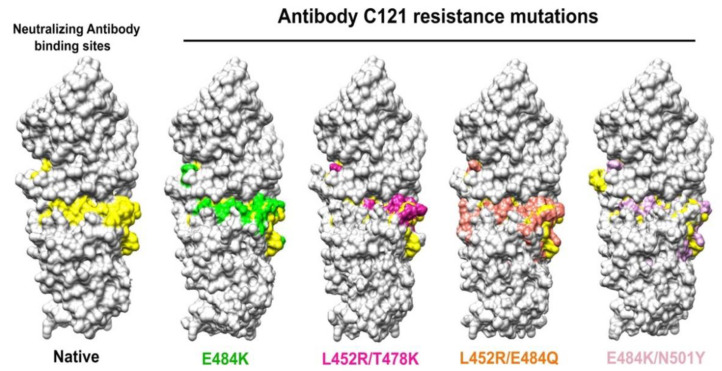
The change in antibody binding sites in the double mutants L452R/T478K (Delta), L452R/E484Q (kappa) and E484K/N501Y (Beta) compared with native antibody binding sites.

**Table 1 membranes-12-00844-t001:** Atomistic and coarse-grained forcefield in MD Simulations.

No.	Forcefield	Drugs	Lipid	DNA & RNA	Protein
1	GROMOS	GROMOS 43A1, GROMOS 45A3/4, GROMOS53A5/6, GROMOS54A7, GROMOS54B7, GROMOS54A8			
2	OPLS	OPLS-AA	OPLS-AA	OPLS-AA/M	OPLS-AA, OPLS-AA/L
3	CHARMM	CHARMM general force field (CGenFF)	CHARMM27 lipids, CHARMM36 lipids	CHARMM27 DNA, CHARMM27 RNA/DNA, CHARMM 36 RNA, CHARMM 36 DNA	CHARMM22/CMAP, CHARM27, CHARMM36, CHARMM36m
4	AMBER	General AMBER force field (GAFF)	LIPID14, LIPID21	AMBER99 OL3, AMBER99bsc, AMBER OL15	AMBER94, AMBER96, AMBER99, AMBER99sb, AMBER03, AMBER14sb, AMBER15ipq, AMBER19sb
5	MARTINI	MARTINI 2, MARTINI22, MARTINI22p, MARTINI 3, MARTINI dry, MARTINI ELNEDYN22, MARTINI ELNEDYNP22	MARTINI 2, MARTINI22, MARTINI22p, MARTINI 3, MARTINI-Dry, MARTINI ELNEDYN22, MARTINI ELNEDYNP22	MARTINI 2015	MARTINI 2, MARTINI22, MARTINI22p, MARTINI 3, MARTINI dry, MARTINI ELNEDYN22, MARTINI ELNEDYNP22
6	Coarse-grained forcefield models (additional)	-	Electrostatics-based model (ELBA) [90] protein-lipid CG model [91]	PRIMONA, DMD, NAST, ENMs, oxRNA, SimRNA, SPQR	Rosetta centroid (CEN), UNRES, CABS, PRIMO, AWSEM, SURPASS, Scorpion, OPEP

## Data Availability

Not applicable.

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
