# Peer review of "Applications of Molecular Dynamics Simulation in Protein Study"

_membranes, 2022, doi:10.3390/membranes12090844_

Round 1
Reviewer 1 Report
Interesting MDS review worth publishing. Just a few comments:
Page 3
line 83:
Amber is also worth mentioning
https://ambermd.org/
AmberTools22 is free of charge:
https://ambermd.org/AmberTools.php
D.A. Case, T.E. Cheatham, III, T. Darden, H. Gohlke, R. Luo, K.M. Merz, Jr., A. Onufriev, C. Simmerling, B. Wang and R. Woods. The Amber biomolecular simulation programs. J. Computat. Chem. 26, 1668-1688 (2005).
line 105:
is "thermos-dynamical"
should be:
"thermo-dynamical"
Pages 3- 4:
Some update is required:
other coarse-grained are also available not only MARTINI:
By the way, the first model that used the concept of a coarse grain model was published in 1975 by Levitt and Warshel (Levitt, M.; Warshel, A. Computer-Simulation of Protein Folding. Nature 1975, 253, 694−698.)
Examples of coarse-grained protein models:
model of Levitt and Warshel 1975 (Levitt, M.; Warshel, A. Computer-Simulation of Protein Folding. Nature 1975, 253, 694−698.)
SICHO (SIde CHain Only) 1998 (Kolinski, A.; Jaroszewski, L.; Rotkiewicz, P.; Skolnick, J. An efficient Monte Carlo model of protein chains. Modeling the short- range correlations between side group centers of mass. J. Phys. Chem. B 1998, 102, 4628−4637.)
Rosetta centroid mode 2004 (Rohl, C.; Strauss, C.; Misura, K.; Baker, D. In Numerical Computer Methods, Part D; Elsevier: Amsterdam, 2004; Vol. 383.)
CABS (C-Alpha, Beta and Side chain) 2004 [Kolinski, A. Protein modeling and structure prediction with a reduced representation. Acta Biochim. Polym. 2004, 51, 349−371.]
UNRES (UNited RESidues) 2014 (Liwo, A.; Baranowski, M.; Czaplewski, C.; Golas, E.; He, Y.; Jagiela, D.; Krupa, P.; Maciejczyk, M.; Makowski, M.; Mozolewska, M. A.; et al. A unified coarse-grained model of biological macromolecules based on mean-field multipole-multipole interactions. J. Mol. Model. 2014, 20, 2306.
SURPASS (Single United Residue per Pre-Averaged Secondary Structure fragment) 2017 (Dawid, A.E.; Gront, D.; Kolinski, A., SURPASS Low-Resolution Coarse-Grained Protein Modeling ,J. Chem. Theory Comput. 2017, 13, 5766−5779)
Examples of coarse-grained nucleic acid models:
DMD (Discrete Molecular Dynamics) 2008 (Ding F.; Sharma S.; Chalasani P.; Demidov V. V.; Broude N. E.; Dokholyan N. V. Ab Initio RNA Folding by Discrete Molecular Dynamics: From Structure Prediction to Folding Mechanisms. RNA 2008, 14, 1164)
Vfold 2009 (Cao S.; Chen S.-J. Predicting Structures and Stabilities for H-type Pseudoknots with Interhelix Loops. RNA 2009, 15, 696)
NAST (The Nucleic Acid Simulation Tool) 2009 (Jonikas M. A.; Radmer R. J.; Laederach A.; Das R.; Pearlman S.; Herschlag D.; Altman R. B. Coarse-grained Modeling of Large RNA Molecules with Knowledge-based Potentials and Structural Filters. RNA 2009, 15, 189)
ENMs (Elastic Network Models) 2013 (Setny P.; Zacharias M. Elastic Network Models of Nucleic Acids Flexibility. J. Chem. Theory Comput. 2013, 9, 5460)
oxRNA 2014 (Sulc P.; Romano F.; Ouldridge T. E.; Doye J. P.; Louis A. A. A Nucleotide-level Coarse-grained Model of RNA. J. Chem. Phys. 2014, 140)
MARTINI 2015 (Uusitalo J. J.; Ingólfsson H. I.; Akhshi P.; Tieleman D. P.; Marrink S. J. Martini Coarse-Grained Force Field: Extension to DNA. J. Chem. Theory Comput. 2015, 11, 3932)
SimRNA 2016 (Boniecki M. J.; Lach G.; Dawson W. K.; Tomala K.; Lukasz P.; Soltysinski T.; Rother K. M.; Bujnicki J. M. SimRNA: A Coarse-grained Method for RNA Folding Simulations and 3D)
SPQR 2017 (Poblete S.; Bottaro S.; Bussi G. A Nucleobase-centered Coarse-grained Representation for Structure)
Page 9 lines 330-331:
This sentence would be improved grammatically: "The improvement can achieve through improving the FF or the method of simulations."
Author Response
Point-by-Point Response to the Reviewers’ Comments
Reviewer #1:
Comments to the Author
Interesting MDS review worth publishing. Just a few comments:
[response] We thank the reviewer for the appreciation of our work and providing comments and suggestions for improving our manuscript. We have now carried out thorough revision of our manuscript as per all the comments and suggestions.
Comment 1:
Page 3; line 83:
Amber is also worth mentioning: https://ambermd.org/;
AmberTools22 is free of charge: https://ambermd.org/AmberTools.php
D.A. Case, T.E. Cheatham, III, T. Darden, H. Gohlke, R. Luo, K.M. Merz, Jr., A. Onufriev, C. Simmerling, B. Wang and R. Woods. The Amber biomolecular simulation programs. J. Computat. Chem. 26, 1668-1688 (2005).
[response] As suggested by reviewer, we have now included the AmberTools22 for Molecular Simulations along with the suggested reference in the revised manuscript.
Comment 2:
line 105: is "thermos-dynamical" should be: "thermo-dynamical".
[response] As pointed out by reviewer, we have now corrected it to thermo-dynamical in the revised manuscript.
Comment 3:
Pages 3- 4; Some update is required: other coarse-grained are also available not only MARTINI:
By the way, the first model that used the concept of a coarse grain model was published in 1975 by Levitt and Warshel (Levitt, M.; Warshel, A. Computer-Simulation of Protein Folding. Nature 1975, 253, 694−698.)
Examples of coarse-grained protein models:
SICHO (SIde CHain Only) 1998 (Kolinski, A.; Jaroszewski, L.; Rotkiewicz, P.; Skolnick, J. An efficient Monte Carlo model of protein chains. Modeling the short- range correlations between side group centers of mass. J. Phys. Chem. B 1998, 102, 4628−4637.)
Rosetta centroid mode 2004 (Rohl, C.; Strauss, C.; Misura, K.; Baker, D. In Numerical Computer Methods, Part D; Elsevier: Amsterdam, 2004; Vol. 383.)
CABS (C-Alpha, Beta and Side chain) 2004 [Kolinski, A. Protein modeling and structure prediction with a reduced representation. Acta Biochim. Polym. 2004, 51, 349−371.]
UNRES (UNited RESidues) 2014 (Liwo, A.; Baranowski, M.; Czaplewski, C.; Golas, E.; He, Y.; Jagiela, D.; Krupa, P.; Maciejczyk, M.; Makowski, M.; Mozolewska, M. A.; et al. A unified coarse-grained model of biological macromolecules based on mean-field multipole-multipole interactions. J. Mol. Model. 2014, 20, 2306.
SURPASS (Single United Residue per Pre-Averaged Secondary Structure fragment) 2017 (Dawid, A.E.; Gront, D.; Kolinski, A., SURPASS Low-Resolution Coarse-Grained Protein Modeling ,J. Chem. Theory Comput. 2017, 13, 5766−5779)
Examples of coarse-grained nucleic acid models:
DMD (Discrete Molecular Dynamics) 2008 (Ding F.; Sharma S.; Chalasani P.; Demidov V. V.; Broude N. E.; Dokholyan N. V. Ab Initio RNA Folding by Discrete Molecular Dynamics: From Structure Prediction to Folding Mechanisms. RNA 2008, 14, 1164)
Vfold 2009 (Cao S.; Chen S.-J. Predicting Structures and Stabilities for H-type Pseudoknots with Interhelix Loops. RNA 2009, 15, 696)
NAST (The Nucleic Acid Simulation Tool) 2009 (Jonikas M. A.; Radmer R. J.; Laederach A.; Das R.; Pearlman S.; Herschlag D.; Altman R. B. Coarse-grained Modeling of Large RNA Molecules with Knowledge-based Potentials and Structural Filters. RNA 2009, 15, 189)
ENMs (Elastic Network Models) 2013 (Setny P.; Zacharias M. Elastic Network Models of Nucleic Acids Flexibility. J. Chem. Theory Comput. 2013, 9, 5460)
oxRNA 2014 (Sulc P.; Romano F.; Ouldridge T. E.; Doye J. P.; Louis A. A. A Nucleotide-level Coarse-grained Model of RNA. J. Chem. Phys. 2014, 140)
MARTINI 2015 (Uusitalo J. J.; Ingólfsson H. I.; Akhshi P.; Tieleman D. P.; Marrink S. J. Martini Coarse-Grained Force Field: Extension to DNA. J. Chem. Theory Comput. 2015, 11, 3932)
SimRNA 2016 (Boniecki M. J.; Lach G.; Dawson W. K.; Tomala K.; Lukasz P.; Soltysinski T.; Rother K. M.; Bujnicki J. M. SimRNA: A Coarse-grained Method for RNA Folding Simulations and 3D)
SPQR 2017 (Poblete S.; Bottaro S.; Bussi G. A Nucleobase-centered Coarse-grained Representation for Structure)
[response] As suggested by the reviewer, we have now updated main text for coarse grain protein and nucleic acid models along with all suggested references in the revised manuscript and Table 1.
Comment 4:
Page 9 lines 330-331:
This sentence would be improved grammatically: "The improvement can achieve through improving the FF or the method of simulations."
[response] As pointed out by reviewer, we have now corrected the sentence in the revised manuscript.

Reviewer 2 Report
literature review can be slightly improved by incorporating MD studies on excited states like:
Ben-Nun, M., Quenneville, J., & Martínez, T. J. (2000). Ab initio multiple spawning: Photochemistry from first principles quantum molecular dynamics. The Journal of Physical Chemistry A, 104(22), 5161-5175.
Jones, C. M., List, N. H., & Martínez, T. J. (2022). Steric and Electronic Origins of Fluorescence in GFP and GFP-like Proteins. Journal of the American Chemical Society.
Wasif Baig, M., Pederzoli, M., Kyvala, M., Cwiklik, L., & Pittner, J. (2021). Theoretical Investigation of the Effect of Alkylation and Bromination on Intersystem Crossing in BODIPY-Based Photosensitizers. The Journal of Physical Chemistry B, 125(42), 11617-11627.
Author Response
Reviewer #2:
Comment 1:
literature review can be slightly improved by incorporating MD studies on excited states like:
- Ben-Nun, M., Quenneville, J., & Martínez, T. J. (2000). Ab initio multiple spawning: Photochemistry from first principles quantum molecular dynamics.The Journal of Physical Chemistry A, 104(22), 5161-5175.
- Jones, C. M., List, N. H., & Martínez, T. J. (2022). Steric and Electronic Origins of Fluorescence in GFP and GFP-like Proteins.Journal of the American Chemical Society.
- Wasif Baig, M., Pederzoli, M., Kyvala, M., Cwiklik, L., & Pittner, J. (2021). Theoretical Investigation of the Effect of Alkylation and Bromination on Intersystem Crossing in BODIPY-Based Photosensitizers.The Journal of Physical Chemistry B, 125(42), 11617-11627.
[response] As suggested by reviewer, we have now updated the MD studies on excited states and suggested references in the revised manuscript.
